# Effect of Rain-Shelter Cultivation on Yield and Fruit Quality of Container-Grown Rabbiteye Blueberry in Central-Eastern China

**DOI:** 10.3390/plants14081167

**Published:** 2025-04-09

**Authors:** Jiguang Wei, Jiafeng Jiang, Liangliang Tian, Yanqin Jiang, Chunfeng Ge, Hong Yu, Qilong Zeng

**Affiliations:** Jiangsu Key Laboratory for the Research and Utilization of Plant Resources, Institute of Botany, Jiangsu Province and Chinese Academy of Sciences (Nanjing Botanical Garden Mem. Sun Yat-Sen), Nanjing 210014, China; weijiguang@jib.ac.cn (J.W.); jiangjiafeng@jib.ac.cn (J.J.); tianliangliang@jib.ac.cn (L.T.); jiangyanqin@jib.ac.cn (Y.J.); gechunfeng@jib.ac.cn (C.G.); yuhong@jib.ac.cn (H.Y.)

**Keywords:** *Vaccinium virgatum* Aiton, gas exchange, open-field cultivation, light conditions, fruit cracking rate

## Abstract

The fruit ripening season for the rabbiteye blueberry often coincides with periods of heavy rainfall in central–eastern China. The use of rain shelters to protect fruit from rainfall damage has increased worldwide due to global climate anomalies. However, the effects of rain-shelter cultivation on the photosynthesis and fruit characteristics of the rabbiteye blueberry have not yet been fully explored. In the present study, 4-year-old container-grown rabbiteye blueberry plants were covered with polyethylene (PE) film from the berry coloration stage until fruit were harvested for three consecutive years in Nanjing, China. The results showed that rain-shelter cultivation did not affect the air temperature and relative humidity, but significantly reduced the photosynthetically active radiation and UV radiation reaching the canopy zone. However, the rain shelter conditions did not significantly decrease the net photosynthetic rate (Pn), stomatal conductance (Gs), and transpiration rate (E) of the rabbiteye blueberry leaves. Additionally, the fruit yield and berry weight of blueberries cultivated under the rain shelter were not significantly affected. Furthermore, no significant differences were observed in total soluble solids, acidity, and total flavonoids content between fruits grown under the rain shelter and in the open field in all experimental years, but a significant decrease in total polyphenols and anthocyanins content was observed in fruits grown under the rain shelter in years with less rainfall. Our results suggest that simple rain shelter cultivation did not noticeably affect the photosynthesis and fruit yield of container-grown rabbiteye blueberry in rainy areas of central–eastern China, but its effects on the fruit quality vary depending on rainfall during the fruit ripening period.

## 1. Introduction

The rabbiteye blueberry (*Vaccinium virgatum* Aiton) is suitable for cultivation in central–eastern China due to the warm climate and large area of acidic soils. One of the most widely planted cultivars is ‘Brightwell’, characterized by its strong adaptability, high yield, and good fruit quality [1]. The two main problems affecting rabbiteye blueberry cultivation in this region are heavy clay soils and fruit ripening that coincides with the rainy season. Heavy clay soils not only lead to high soil amendment costs, but also result in poor drainage during the rainy season [2,3,4]. Heavy rainfall during the fruit ripening period not only causes fruit drop and cracking [5], but also adversely affects fruit harvesting.

Innovative production systems have been explored due to the high soil amendment costs and long periods of negative cash flow in traditional soil cultivation [6]. Container-based production enables precise control of water and nutrient levels in the rhizosphere and allows growers to cultivate blueberries on land with soil problems such as soil salinity, infertility, and poor drainage [7,8,9].

Rain shelter cultivation is widely used for fruit production in rainy regions. However, shelter covering modifies micrometeorological factors, such as light intensity, air temperature, and humidity, which could affect the physiological performance and plant growth of the trees, thereby affecting fruit yield and quality. Many studies have shown that rain shelter cultivation is beneficial in preventing fruit from cracking and clearly improves yield and fruit quality in several fruits [10,11,12,13,14]. Due to differences in the structure and covering material used, some studies have found that rain-shelter cultivation has no positive effect or a negative effect on tree physiology, plant growth, fruit yield, or quality. Zhang et al. [15] found that the net photosynthetic rate (Pn) in the leaves of sweet cherry plants was significantly reduced due to the insufficient light provided by the rain shelter with approximately 55% transmittance. Pino et al. [16] reported that the plastic rain cover caused an increase in air temperature and a decrease in relative humidity, while the environmental changes observed under the rain cover did not affect the CO_2_ assimilation, but tended to reduce the soluble solids content, anthocyanins, total phenolics, and carotenoid contents in the fruit of sweet cherry plants. Similar findings were reported by Meng et al. [17] in a 2-year investigation of grape cultivation under a rain shelter. In blueberries, it has been demonstrated that the yields of ‘Toro’, ‘Nui’, ‘Legacy’, and ‘Misty’ were significantly enhanced when grown under plastic tunnels [18]. Similarly, Retamal-Salgado et al. [19] observed that blueberry production under high tunnels can improve fruit precocity and increase fruit yield due to higher temperatures and diffuse photosynthetically active radiation. These results differ from those obtained by Ogden and van Iersel [20], who indicated that southern highbush blueberry production in high tunnels affected the synchronization of flowering and pollination, leading to a reduction in fruit set and yield. Matamala et al. [21] also reported that the type of cover differentially affects yield and fruit quality in blueberries due to the specific light and temperature conditions generated under these materials. Therefore, previous findings suggest that the impact of shelter covering on the yield and fruit quality of highbush blueberries varies depending on local climate, shelter types, cover material, and cultivar. However, there is little information regarding the effects of shelter covering on tree physiology and fruit characteristics of rabbiteye blueberries.

The objective of the present research was to investigate the effect of rain cover on the microclimate, plant physiology, fruit yield, and quality of the rabbiteye blueberry, ‘Brightwell’, grown in central–eastern China.

## 2. Results

### 2.1. Soil Moisture Conditions

As observed in most periods during the film covering in 2020, the soil moisture content under the rain-shelter cultivation was lower than that in the open-field cultivation, but the difference was not statistically significant (Figure 1).

### 2.2. Photosynthetic Characteristics

Compared with leaves of plants cultivated in the open field, the net photosynthetic rate (Pn), transpiration rate (E), stomatal conductance (Gs), and the intercellular CO_2_ concentration (Ci) in the leaves of plants cultivated under the rain shelter were not significantly decreased in the month when the rain cover was installed (July), or in the month before the rain cover was installed (May), or in the month after the rain cover was removed (September) (Figure 2).

### 2.3. Fruit Drop and Cracking Rate

Shelter covering significantly reduced the fruit drop and cracking rates of the blueberries in all 3 years of the study (Figure 3). Additionally, the fruit drop and cracking rates were significantly higher in years with more precipitation (2020 and 2021) than in years with less precipitation (2019). Moreover, the interaction effect of treatments and experimental years was significant (*p* < 0.05).

### 2.4. Fruit Weight and Yield

As the years of planting increased, the fruit yield showed a significant downward trend. However, no significant differences were observed in fruit yield between trees grown under the rain shelter and in the open field in all experimental years. The fruit yield under rain-shelter cultivation was 1.6% lower than that of the control treatment in years with less precipitation (2019), while the fruit yield from the sheltered trees was 1.7% and 1.4% higher in 2020 and 2021, respectively, than those from the open-field trees (Figure 4A). There was no significant interaction between treatment and experimental years (*p* > 0.05).

In the three experimental years, the berry weight of the fruit grown under the rain-shelter treatment was lower than that of the open-field treatment; however, there was no significant difference between the two treatments (Figure 4B). No significant interaction was found between treatments and experimental years in berry weight (*p* > 0.05).

### 2.5. Fruit Quality

In 2019, trees grown in open-field conditions had significantly higher total polyphenols (4.52 ± 0.32 GAE mg/g) and total anthocyanins (209.51 ± 12.18 C3GE mg/100 g) compared to those grown under rain-shelter cultivation (4.09 ± 0.23 GAE mg/g and 185.09 ± 8.76 C3GE mg/100 g, respectively). However, there were no significant differences in total soluble solids and total flavonoids between the two treatments in 2019. In 2020 and 2021, no significant differences were observed between the two cultivation modes for any of the measured parameters, including total soluble solids, acidity, total polyphenols, total flavonoids, and total anthocyanins (Table 1). Additionally, there was no significant interaction between treatments and experimental years (*p* > 0.05).

## 3. Discussion

### 3.1. Effects of Shelter Covering on the Micrometeorological Conditions

Shelter covering has been confirmed to alter the microclimate around the plant canopy, with changes in air temperature, relative humidity, wind speed, and light intensity. These effects are mainly dependent on the rain shelter structure and covering materials [19,22,23,24]. The simple rain shelter used in the present study is a roof-type structure covered with a 1.8 m wide plastic film on the top and there is no covering on the sides. Therefore, there was no significant difference in the average daily air temperature and relative humidity between the rain-shelter and open-field treatments. However, the photosynthetically active radiation and ultraviolet radiation under the rain shelter were significantly decreased due to the radiation being intercepted by the plastic film covering.

### 3.2. Effects of Shelter Covering on the Photosynthetic Characteristics of Rabbiteye Blueberry Leaves

Shelter covering unavoidably decreases the radiation reaching the crop canopy, thus presumably leading to a reduction in photosynthetic capacity [15]. Depending on the cultivar and growing stage, the saturating light level for the photosynthesis of the blueberries was estimated to be 600 to 800 μmol·m^−2^·s^−1^ [25,26,27,28,29]. In the present study, the net photosynthetic rate tended to be lower in the leaves of blueberries under the rain shelter, which was largely due to lower irradiation under the rain shelter with 73% transmittance. However, the light intensity under the rain shelter was still higher than the light saturation point of the blueberries, which ensured that the blueberry leaves could achieve the highest photosynthesis for adequate growth and development. Therefore, there was no significant difference in the photosynthetic rate between the two treatments. Additionally, the low light environment may cause the stomata to close or remain partially closed, lowering stomatal conductance [30], which thereby contributed to the decrease in transpiration rate of blueberry plants cultivated under rain shelter cultivation (Figure 2). However, it is worth mentioning that there was also no significant difference in these photosynthetic parameters between the two cultivation modes. This indicated that the synthesis and accumulation of photosynthetic products are basically similar between the two treatments.

### 3.3. Effects of Shelter Covering on the Fruit Yield of Rabbiteye Blueberry

Rain shelters have been proven to be a preferable means of protecting fruit against heavy rain episodes during the period of maturity, thereby significantly reducing the fruit drop and cracking rates, and increasing fruit yields [13,31,32,33]. Our results also showed that rain shelters markedly reduced the fruit drop and cracking rate (Figure 3) Since no significant differences were observed in soil moisture content between the two treatments, the higher rate of fruit drop and cracking in open-field cultivation may be attributed to the higher probability of fruit being exposed to rain [34]. This was also supported by the fact that the fruit drop and cracking rates were significantly higher in years with more precipitation (2020 and 2021) than in years with less precipitation (2019).

In the present study, no significant differences were observed in fruit yield or berry weight between trees grown under the rain shelter and those grown in open-field conditions in all experimental years. Possible reasons for this include the following: (1) Given the extremely low rate of rain-induced fruit drop and cracking in all experimental years, their impact on yield can be considered negligible. (2) Although the light intensity under the rain shelter was significantly lower than that in the open-field control treatment, no significant differences were observed in the photosynthetic capacity of the blueberry leaves between the two treatments.

### 3.4. Effects of Shelter Covering on the Fruit Quality of Rabbiteye Blueberry

Most related studies have shown that rain-shelter cultivation increases the soluble solids content of fruits [10,13,24,35], while other studies indicated that the soluble solids content varied depending on the fruit cultivars [16]. However, the results of this study showed that no significant differences were noted in the total soluble solids content between fruits grown under the rain shelter and those grown in open-field conditions in the three experimental years. Similar findings were reported by Gao et al. [23] in a 2-year investigation of grape cultivation under rain-shelter conditions. Our present study revealed that rain-shelter cultivation did not affect the acidity content of the rabbiteye blueberry fruits. Previous studies have also reported similar results in grape [23] and cherry [13]. It was speculated that shelter covering had no detectable effect on the other meteorological factors except for light intensity, and there was no significant difference in the photosynthetic assimilation ability between the rain-shelter treatment and the open-field control. Therefore, the synthesis and accumulation of sugars and acids are basically the same in the two cultivation modes.

In the present study, the total polyphenols and anthocyanins contents were significantly lower in blueberry fruits grown under rain-shelter conditions in the year with less precipitation (2019), which might be a consequence of the reduction in solar radiation under the rain shelter being more obvious in sunny days than in rainy days or cloudy days. These results aligned with previous studies indicating that higher solar radiation benefited the synthesis of these compounds [16,17]. However, the opposite results were reported by Schmitz-Eiberger and Blanke [36] and Tian et al. [13], which was probably attributable to the increase in temperature under the shelter [37]. Similar findings were observed by Li et al. [10], who reported that rain-shelter cultivation increased the concentrations of almost all the anthocyanin components detected in the ripening grape berries. This promotion was possibly achieved by enhancing the supply of photosynthetic assimilates from the grape leaves since the senescence of the leaves under the rain-shelter cultivation was delayed. In contrast, during 2020 and 2021, when precipitation was higher, no significant differences were found between the two treatments for either the total polyphenols or the anthocyanin content. This indicates that higher rainfall may mitigate the differences in fruit quality between open-field and rain-shelter cultivation.

Overall, the results suggest that the impact of rain-shelter cultivation on rabbiteye blueberry fruit quality is influenced by the amount of rainfall during the maturation period. Lower rainfall years may amplify the differences between cultivation methods, while higher rainfall years may result in more uniform fruit quality regardless of the cultivation methods.

## 4. Materials and Methods

### 4.1. Experimental Site

The experimental field was located at the Institute of Botany, Jiangsu Province and Chinese Academy of Sciences (32°3′29″ N, 118°49′46″ E), Nanjing, China. The experimental area is characteristic of a typical subtropical monsoon climate with an average annual air temperature of 15.7 °C and mean annual precipitation of 1021 mm. The precipitation is mainly concentrated from June to September. The average frost-free period is about 237 days.

### 4.2. Plant Cultivation and Experimental Design

On 17 March 2016, 6-month-old cuttings of ‘Brightwell’ rabbiteye blueberry (*V. virgatum*) were transplanted into 15 L plastic pots, with one plant per pot.

Each pot was filled with a mixture of soil, peat, and perlite in a ratio of 60%:20%:20% (by volume), which is widely used in current practices by blueberry growers in southern China. Plants were grown in the open field and were watered about twice a week to maintain soil moisture near field capacity. A dry-blend (15N-15P-15K) fertilizer was applied at 10 g per plant once a month, from May to October 2016. In the years thereafter, the fertilizer rate and regime were adjusted according to plant growth stages following commercial practices.

The experiment was carried out from 2019 to 2021. Thirty-six similarly growing plants were selected and arranged in six rows. The experimental plants were spaced 1.2 m within rows and 2.5 m between rows. Three rows were chosen for the open-field treatment (control), and the remaining three rows were used for the rain-shelter treatment. Each treatment was replicated three times with six plants per replicate.

The rain-shelter frame was designed as a roof-type structure with a 6 m length, a 1.8 m width, and a 2.2 m height above the ground (Figure 5). The shelters were covered with a colorless polyethylene (PE) film with a thickness of 0.10–0.12 mm and 75% light transmittance. The PE film was set up before berry coloration in early June and removed after fruit harvest.

### 4.3. Field Management

During the experimental period, the blueberry plants were drip irrigated by an automatic irrigation timer, with one dripper (2 L·h^−1^) per pot. Irrigation was applied in four cycles daily from 06:00 to 15:00 at 3 h intervals. The duration of each cycle was adjusted throughout the growing season to maintain drainage at ≥20% of total water input to ensure that plants grew normally. Plants grown in the open field were not irrigated on days with ≥5 mm rainfall. A dry-blend (15N-15P-15K) fertilizer was applied monthly at 30 g per plant, from March to October in each year. Pruning, weed control, and pest and disease management followed standard commercial practices

### 4.4. Environmental Conditions

To understand the changes in environmental conditions under the rain-shelter and the open-field treatments, environmental parameters were monitored during film covering period. The air temperature and relative humidity (RH) were measured by using automatic sensors (ZDR-20; Zeda Instruments Co., Ltd., Hangzhou, China) at 30 min intervals. The sensors were installed in the center of the experimental row and at 1.8 m height. The photosynthetically active radiation (PAR) and UV radiation measurements were taken at 09:00, 11:00, 13:00, 15:00 and 17:00 on sunny and cloudy days. The PAR was measured using a quantum sensor (LI-250A; Li-Cor, Lincoln, NE, USA). UV-A and UV-B radiation were measured by using ultraviolet radiometer (UV-A, UV-B; Photoelectric Instrument Factory of Beijing Normal University, Beijing, China). Precipitation data were obtained from an automatic weather station (NL-5G; Zhejiang Top Cloud-agri Technology Co., Ltd., Hangzhou, China) located in the experimental field. No significant difference was observed in the average daily air temperature and relative humidity between the rain-shelter and open-field treatments (Figure 6). Compared with the open field, the PAR, UV-A radiation, UV-B radiation under the rain-shelter treatment was dramatically reduced by 27%, 32%, 35%, respectively, on sunny days. Similarly, the radiation levels under the rain shelter were significantly lower on cloudy days than those in the open field (Table 2). As shown in Figure 2, the total precipitation was 182.4 mm, 718.8 mm, and 485.2 mm during the film-covering period in 2019, 2020, and 2021, respectively (Figure 7).

### 4.5. Soil Moisture Content Measurement

Soil moisture content in pots was measured once a week during the 2020 film covering period using a portable TDR soil moisture sensor (WET-2, Delta-T Devices Ltd., Burwell, UK).

### 4.6. Photosynthetic Characteristics Measurement

In 2020, photosynthetic activity was determined on fully expanded leaves from 09:30 to 11:00 on clear days using a portable photosynthesis system (LI-6800F; Li-Cor, Inc., Lincoln, NE, USA). The photosynthetically active radiation (PAR) level in the leaf chamber was set to 1200 µmol·m^−2^·s^−1^ and the reference CO_2_ concentration was maintained at 400 μmol·mol^−1^ during the measurements.

### 4.7. Fruit Yield and Berry Quality Measurements

The three plants in the middle of each row were used for data collection to avoid border effects. In total, nine plants (three plants per replicate and three replicates per treatment) were selected for yield and berry quality determinations. Ripe fruit from each plant was harvested and weighed weekly during the harvest season each year to determine total berry yield per plant.

During peak production time, 60 berries from each plant were randomly selected to estimate average single-berry weight. In 2019, the fruit’s soluble solid content (SSC) was determined using ten ripe berries from each plant with a refractometer (WYT; Optical Instrument Co., Ltd., Chengdu, China). In 2020–2021, SSC (expressed as °Brix) and acidity were measured using a pocket Brix-Acidity meter (PAL-BIXACID F5, Atago Co., Ltd., Tokyo, Japan).

Then, 50 g of ripe berries from each plant were ground into a homogenate and 2.5 g of homogenate was mixed with 15 mL acidified ethanol (60% EtOH: 0.1M HCl, 85:15 *v*/*v*), and extracted using ultrasonic-assisted extraction for 20 min at room temperature. The extract was centrifuged at 4000 rpm for 15 min. The supernatant was collected and the residual tissue was re-extracted two additional times under the same conditions. The three obtained supernatants were combined, diluted to 50 mL with pH 3.0 60% ethanol. The extracts were kept at 4 °C in the dark until analysis.

The total phenolic content was evaluated using the Follin–Ciocalteu colorimetric method [38]. A 0.1 mL sample of the extract was diluted to a final volume of 5 mL with distilled water. The mixture was then oxidized with 0.5 mL Follin–Ciocalteu reagent. After 5 min, the reaction mixture was neutralized by 1.5 mL 75 g·L^−1^ Na_2_CO_3_. The absorbance was measured at 760 nm after incubation for 90 min at room temperature in the dark. Using a standard curve prepared with gallic acid, the total phenolic content was calculated and expressed as milligram of gallic acid equivalent (GAE) per gram of fresh weight (FW).

The total flavonoid content was measured using a colorimetric method [39]. A 1 mL sample of the extract was mixed with 1.4 mL of distilled water. Then, 0.3 mL of 5% NaNO_2_ was added and 0.3 mL of 10% AlCl_3_ was added 5 min later. After 6 min, 2 mL of 1 M NaOH was added and allowed to stand for 15 min. The reaction solution was mixed thoroughly, and the absorbance was measured at 510 nm. The total flavonoid content was calculated and expressed as milligram of quercetin equivalents (QE) per gram of fresh weight (FW).

The total anthocyanin content of the blueberry extract was determined using the spectrophotometric pH differential method [40]. In brief, 1.0 mL of sample was mixed with 10 mL of 0.025 M potassium chloride (pH 1.0), and the absorbance was measured at 520 nm. A second 1.0 mL aliquot was mixed with 10 mL of 0.4 M sodium acetate (pH 4.5), and the absorbance at 520 nm was measured. The total anthocyanin content was calculated using the following formula:Total anthocyanin content (mg·g^−1^) = (*A* _pH 1.0_ − *A* _pH 4.5_) × *V* × *MW* × *DF*/(*ε* × *l* × *m*)
where *A* _pH 1.0_ is the absorbance of the sample diluted with pH 1.0 buffer solution, *A* _pH 4.5_ is the absorbance of the sample diluted with pH 4.5 buffer solution, *V* is the total volume of the blueberry extract (mL), *DF* is the dilution factor (11), *MW* is the average molecular weight of anthocyanins (433.2 g·mol^−1^), *ε* is the extinction coefficient (31,600 L·mol^−1^·cm^−1^), *l* is the path length (1 cm), and *m* is the fresh weight of fruits used for extraction (g).

### 4.8. Statistical Analysis

The statistical analysis was performed using SPSS 16.0 software (SPSS Inc., Chicago, IL, USA), and significant differences between the treatments were assessed by Duncan’s Multiple Range Test at the 0.05 level. All the presented data are expressed as the mean ± standard deviation (SD).

## 5. Conclusions

In conclusion, the rain-shelter cultivation did not affect the air temperature and relative humidity, but significantly reduced the level of photosynthetically active radiation and UV radiation reaching the canopy zone. However, the rain-shelter conditions did not significantly reduce the photosynthetic capacity of rabbiteye blueberry leaves. Therefore, the fruit yield and berry weight under rain-shelter cultivation were not significantly affected. Additionally, there were no significant differences in fruit quality parameters between these two cultivation treatments, except for a significant decrease in total polyphenols and anthocyanin content in fruits grown under the rain-shelter in years with less precipitation. The results of this study suggested that rain-shelter cultivation does not negatively affect photosynthetic characteristics and fruit yield of rabbiteye blueberries, but it could be detrimental to some quality parameters in years with less rainfall.

## Figures and Tables

**Figure 1 plants-14-01167-f001:**
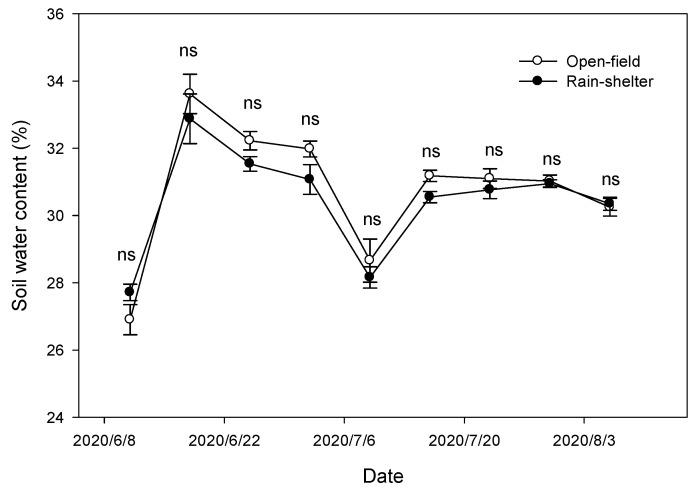
Changes in the soil water content under rain-shelter and open-field cultivation (control) during film covering period in 2020. ns: no significance. Means ± SD (*n* = 3).

**Figure 2 plants-14-01167-f002:**
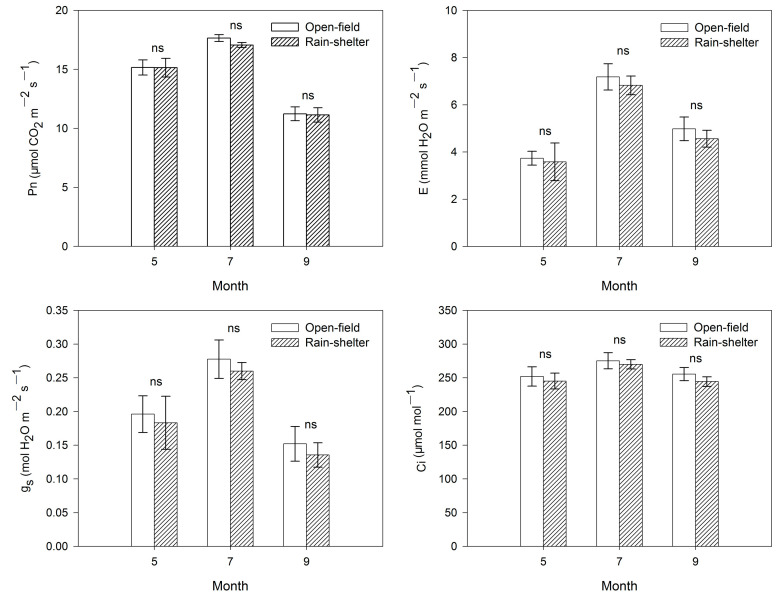
Comparison of the net photosynthetic rate (Pn), transpiration rate (E), stomatal conductance (Gs), and the intercellular CO_2_ concentration (Ci) in leaves of rabbiteye blueberry under rain-shelter and open-field cultivation in 2020. For each month, “ns” indicates no statistical difference. Means ± SD (*n* = 3).

**Figure 3 plants-14-01167-f003:**
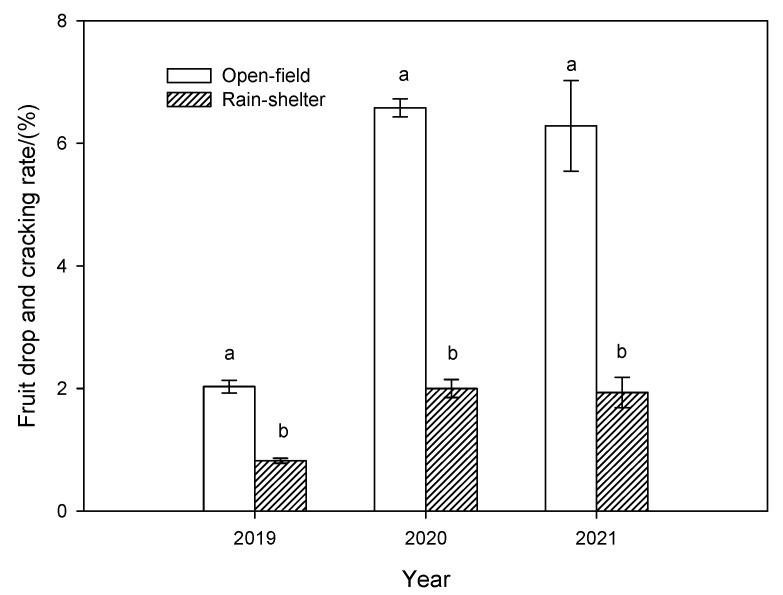
Fruit drop and cracking rates of rabbiteye blueberry under rain-shelter and open-field cultivation in 2019–2021. For each year, different letters indicate significantly different between treatments (*p* < 0.05). Means ± SD (*n* = 3).

**Figure 4 plants-14-01167-f004:**
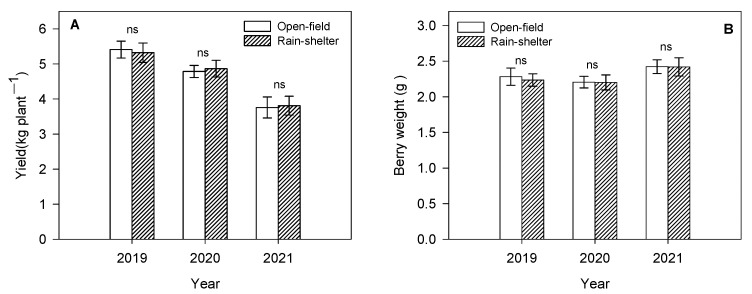
Fruit yield (**A**) and berry weight (**B**) of rabbiteye blueberry under rain-shelter and open-field cultivation in 2019–2021. For each year, “ns” indicates no statistical difference. Means ± SD (*n* = 3).

**Figure 5 plants-14-01167-f005:**
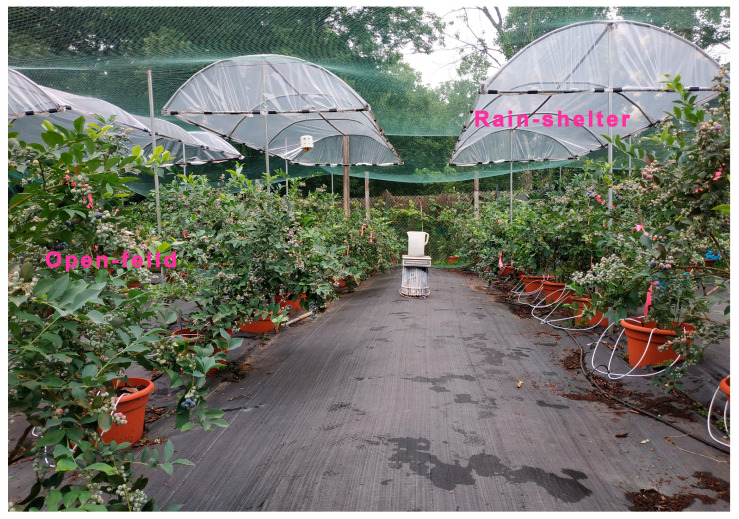
Photos of rabbiteye blueberry cultivated under rain-shelter and open-field conditions.

**Figure 6 plants-14-01167-f006:**
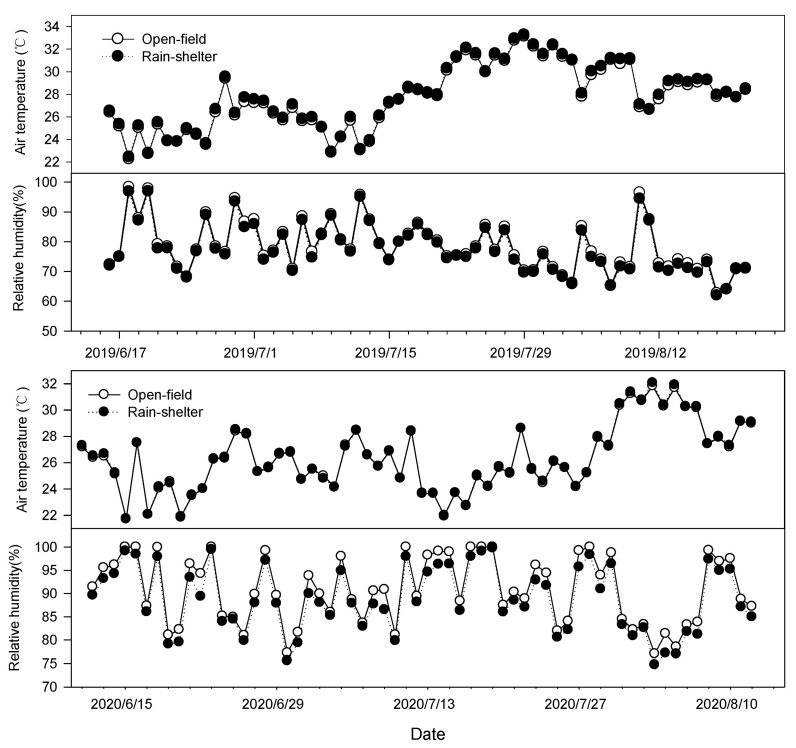
The average daily air temperature and relative humidity under rain-shelter and open-field cultivation (control) during film-covering period in 2019–2020.

**Figure 7 plants-14-01167-f007:**
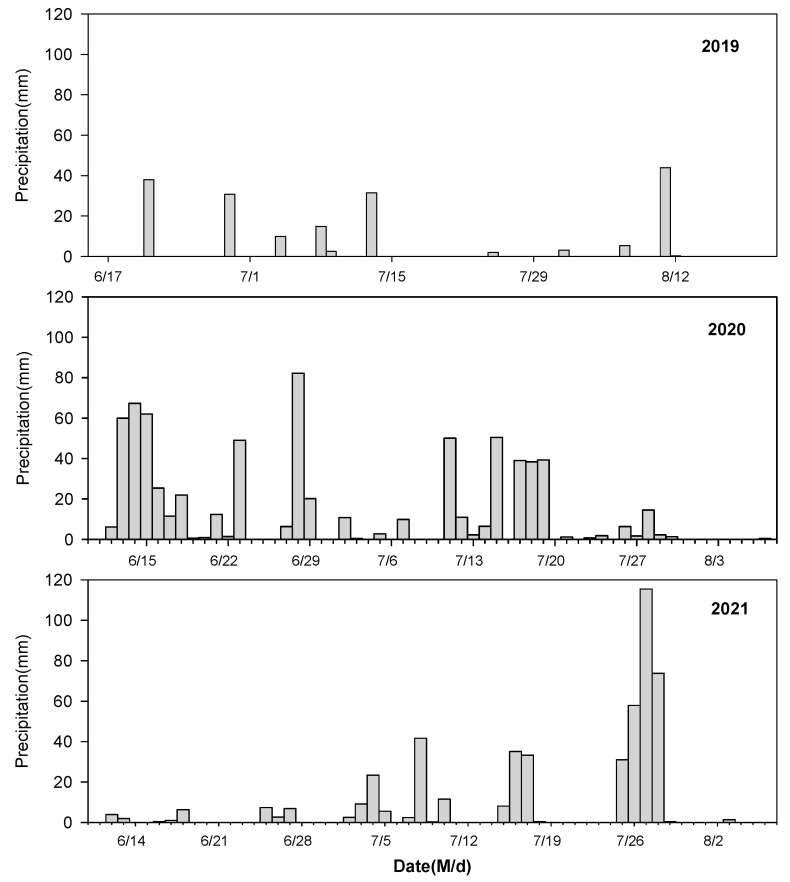
The daily precipitation during film covering period in 2019–2021.

**Table 1 plants-14-01167-t001:** The fruit interior quality of rabbiteye blueberry under rain-shelter and open-field cultivation in 2019–2021.

Year	Treatment	Total Soluble Solids (%)	Acidity (%)	Total Polyphenols GAE mg/g	Total Flavonoids mg QE/g	Total Anthocyanins C3GE mg/100 g
2019	Open field	12.4 ± 0.6 a	-	4.52 ± 0.32 a	3.74 ± 0.40 a	209.51 ± 12.18 a
	Rain shelter	12.2 ± 0.5 a	-	4.09 ± 0.23 b	3.55 ± 0.36 a	185.09 ± 8.76 b
2020	Open field	11.3 ± 0.4 a	0.37 ± 0.03 a	3.46 ± 0.33 a	3.85 ± 0.34 a	171.95 ± 12.60 a
	Rain shelter	11.4 ± 0.6 a	0.38 ± 0.03 a	3.45 ± 0.21 a	3.79 ± 0.28 a	173.35 ± 12.86 a
2021	Open field	11.9 ± 0.4 a	0.44 ± 0.04 a	2.86 ± 0.37 a	3.93 ± 0.25 a	177.82 ± 23.08 a
	Rain shelter	11.9 ± 0.5 a	0.42 ± 0.03 a	2.83 ± 0.31 a	3.96 ± 0.35 a	176.09 ± 21.66 a

Data are presented as means ± SD. For each year, different letters indicate significantly different between treatments (*p* < 0.05).

**Table 2 plants-14-01167-t002:** Photosynthetically active radiation (PAR), UV-A radiation, and UV-B radiation under rain-shelter and open-field cultivation (control) on a sunny day and a cloudy day in 2020.

Weather Condition	Treatment	Photosynthetically Active Radiation (μmol·m^−2^·s^−1^)	UV-A Radiation (W·m^−2^)	UV-B Radiation (W·m^−2^)
Sunny day	Open field	1163.66 ± 21.79 a	12.19 ± 0.16 a	0.88 ± 0.05 a
	Rain shelter	850.49 ± 16.06 b	8.29 ± 0.08 b	0.57 ± 0.01 b
Cloudy day	Open field	234.21 ± 11.05 a	3.00 ± 0.08 a	0.27 ± 0.01 a
	Rain shelter	178.55 ± 10.93 b	2.23 ± 0.04 b	0.20 ± 0.01 b

Data are presented as means ± SD. For each weather condition, means followed by different lowercase letters differ statistically at *p* < 0.05.

## Data Availability

The data presented in this study are available on request from the corresponding author.

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
