# Peer review of "Effect of Rain-Shelter Cultivation on Yield and Fruit Quality of Container-Grown Rabbiteye Blueberry in Central-Eastern China"

_plants, 2025, doi:10.3390/plants14081167_

Round 1
Reviewer 1 Report
Comments and Suggestions for Authors
The manuscript entitled “Effects of rain-shelter cultivation on yield and fruit quality of container-grown rabbiteye blueberry in Central-Eastern China” reports that rain-shelter cultivation reduces fruit drop and cracking without reducing production and affecting physiological and biochemical properties of blueberry fruits, compared to that of open-field cultivation. It is an interesting topic. However, the manuscript has unclear and ambiguous statements that need to be addressed by the authors. Importantly, the results and statistical analysis should be re-considered. Please see my specific comments. Thank you.
Result
Figure 2 is unclear. Figure legends of P and 5 mm line represent for? Please describe the full abbreviation in the Figure caption.
Figure 3. Might be the authors forgot to add small-case letters indicating the statistical differences between open-field and rain-shelter. Or non-significant differences?
Figure 4. How do you manage the statistical analysis for leaf gas exchange parameters?
Lines 124-125: If this statement is a part of figure caption, do not separate it. Combined with Lines 122-123. Please also check the similar mistakes in other figures and tables.
Figures 5-6. In this case, there has only two treatments, open-field and rain-shelter. Just use t-test and compare two treatments on each year. Then, find the main factors and interaction effects of treatments and experimental years.
In all figures, figure captions should be described completely.
Section 2.3 and figure 5: Please use “fruit drop” instead of using “fruit falling”. Refers to previous references.
Table 2. Importantly, there has only two-treatments, open-field and rain-shelter. As mentioned above, the data could be varied depending on the weather and environmental conditions of each year. Therefore, the statistical analysis should be done separately and just compare open-field and rain-shelter. Then, find the main factor and interaction effects of treatment and experimental year.
Discussion
Discussion should be made after revising the statistical analysis and results. Then, discuss properly.
Materials and Methods
Lines 247-250: The pictures of rain-shelter and open-field must be provided to improve the readability of the manuscript.
Author Response
Comments 1: [Figure 2 is unclear. Figure legends of P and 5 mm line represent for? Please describe the full abbreviation in the Figure caption.]
Response 1: [Thank you for pointing this out. Figure legends of P represents precipitation. 5 mm line represent for daily precipitation that was equal to 5 mm. We have deleted them in the revised manuscript. ]
Comments 2: [Figure 3. Might be the authors forgot to add small-case letters indicating the statistical differences between open-field and rain-shelter. Or non-significant differences?]
Response 2: [Yes, there was no significant difference in soil moisture between the rain shelter treatment and the open field treatment.]
Comments 3: [Figure 4. How do you manage the statistical analysis for leaf gas exchange parameters?]
Response 3: [We conducted a significant difference analysis for photosynthetic parameters the open-field and rain-shelter treatments on each month.]
Comments 4: [Lines 124-125: If this statement is a part of figure caption, do not separate it. Combined with Lines 122-123. Please also check the similar mistakes in other figures and tables.]
Response 4: [We agree with your opinion. Therefore we have changed in the revised manuscript.]
Comments 5: [Figures 5-6. In this case, there has only two treatments, open-field and rain-shelter. Just use t-test and compare two treatments on each year. Then, find the main factors and interaction effects of treatments and experimental years.]
Response 5: [We agree with this comment. Therefore we performed one-way ANOVA on two treatments for each year. We also conducted a two-factor ANOVA with treatment and experimental year. The analysis demonstrated significant interaction effects for fruit drop and cracking rates, indicating that treatment efficacy was modulated by inter-annual climatic variability. However, no significant interaction was found between treatment and experimental years in fruit yield and berry weight.]
Comments 6: [In all figures, figure captions should be described completely.]
Response 6: [Thank you for pointing this out. We have already supplemented and improved these in the revised manuscript.]
Comments 7: [Section 2.3 and figure 5: Please use “fruit drop” instead of using “fruit falling”. Refers to previous references.]
Response 7: [We agree with this comment. Therefore, we changed “fruit falling” into “fruit drop” in the revised manuscript.]
Comments 8: [ Table 2. Importantly, there has only two-treatments, open-field and rain-shelter. As mentioned above, the data could be varied depending on the weather and environmental conditions of each year. Therefore, the statistical analysis should be done separately and just compare open-field and rain-shelter. Then, find the main factor and interaction effects of treatment and experimental year.]
Response 8: [Thank you for pointing this out. We carried out one-way ANOVA on two treatments for each year. We also conducted a two-factor ANOVA with treatment and experimental year. The analysis demonstrated there was no significant interaction between treatment and experimental years in any of the fruit interior quality parameters.]
Comments 8: [ Discussion should be made after revising the statistical analysis and results. Then, discuss properly.]
Response 8: [Thank you for your constructive feedback. Following the revisions to the statistical analysis and results section, we have thoroughly updated the Discussion to align with the refined methodology and findings in the revised manuscript.]
Comments 9: [Materials and Methods
Lines 247-250: The pictures of rain-shelter and open-field must be provided to improve the readability of the manuscript.]
Response 9: [We agree with this comment. Therefore, we add pictures in the revised manuscript. ]
Reviewer 2 Report
Comments and Suggestions for Authors
The authors prepared the manuscript on the Effect of rain-shelter cultivation on yield and fruit quality of container-grown rabbiteye blueberry in Central-Eastern China. The aim of the study was to investigate the effect of rain cover on the microclimate, plant physiology, fruit yield and quality in rabbiteye blueberry ‘Brightwell’ grown in Central-Eastern China.
Air temperature and precipitation should not be included as a result of the experiment, but as part of the methodology. Perhaps the data in Table 1 could also be provided in the methodology section.
Lines 122 - 125 make into one paragraph.
Figures add error bars to both sides up and down.
Differences - letters - show significant different in the whole figure? or only between open-field and rain-shelter?
Table 2 What about the differences between years?
So, you made experiment 3 years but photosynthetic measurements were only made for one year? Field experiments require replications. One year of data does not provide stable reliability.
Author Response
Comments 1: [Air temperature and precipitation should not be included as a result of the experiment, but as part of the methodology. Perhaps the data in Table 1 could also be provided in the methodology section.]
Response 1: [Thank you for pointing this out. We agree with this comment. Therefore, we have moved this section to Section 4.4 in the revised manuscript.]
Comments 2: [Lines 122 - 125 make into one paragraph.]
Response 2: [We agree with your opinion. Therefore we have changed in the revised manuscript.]
Comments 3: [Figures add error bars to both sides up and down.]
Response 3: [We agree with this comment. Therefore we have modified the figures in the revised manuscript.]
Comments 4: [Differences - letters - show significant different in the whole figure? or only between open-field and rain-shelter?]
Response 4: [Thank you for pointing this out. we have refined the figures in the revised manuscript. Differences letters show significant different between open-field and rain-shelter.]
Comments 5: [Table 2 What about the differences between years?]
Response 5: [We conducted a two-factor ANOVA with treatment and experimental year. The analysis demonstrated there were significant differences in total soluble solids, titratable acid, total polyphenols, and total anthocyanins between different years. However, no significant interaction between treatment and experimental years was found in any of the fruit interior quality parameters.]
Comments 6: [So, you made experiment 3 years but photosynthetic measurements were only made for one year? Field experiments require replications. One year of data does not provide stable reliability.]
Response 6: [Thank you for pointing this out. We agree that multi-year physiological measurements are ideal, and multi-year data could provide stable reliability. In this study, photosynthetic parameters were indeed only collected during the second year of the experiment due to instrument limitations. Future long-term studies will prioritize continuous physiological monitoring.]
Round 2
Reviewer 1 Report
Comments and Suggestions for Authors
The authors addressed all comments and responded well.
1. Additional suggestion: fruit drop and fruit cracking rates should be presented separately if possible (Figure 3).
2. The manuscript still needs to check some grammatical errors.
Thank you.
The manuscript still needs to check some grammatical errors.
Author Response
Comments 1: [1. Additional suggestion: fruit drop and fruit cracking rates should be presented separately if possible (Figure 3).]
Response 1: [Thank you for your constructive suggestion. Due to the presence of cracked fruits in fallen fruits, it is difficult to distinguish whether they fall first and then crack, or they crack first and then fall to the ground. Therefore, we calculate fallen and cracked fruits together.]
Comments 2: [2. The manuscript still needs to check some grammatical errors.]
Response 2: [Thank you for pointing this out. We have thoroughly reviewed and corrected the grammatical errors in the manuscript, and highlighted them in green font in the revised manuscript.]
Reviewer 2 Report
Comments and Suggestions for Authors
The authors have made many important corrections, and the quality of the manuscript has improved. I have no further comments.
Author Response
Comments 1: [The authors have made many important corrections, and the quality of the manuscript has improved. I have no further comments.]
Response 1: [We sincerely appreciate your constructive feedback and recognition of the revisions made to improve the manuscript. Thank you for your time and expertise in reviewing our paper.]